# The absence of fans removes the home advantage associated with penalties called by National Hockey League referees

**Joël Guérette, Caroline Blais, Daniel Fiset** *

Department of Psychoeducation and Psychology, University of Quebec in Outaouais, Gatineau, Quebec, Canada

* daniel.fiset@uqo.ca

## Abstract

The COVID-19 pandemic has had a major impact on professional sports, notably, forcing the National Hockey League to hold its 2020 playoffs in empty arenas. This provided an unprecedented opportunity to study how crowds may influence penalties awarded by referees in an ecological context. Using data from playoff games played during the COVID-19 pandemic and the previous 5 years ($n = 547$), we estimate the number of penalties called by referees depending on whether or not spectators were present. The results show an interaction between a team's status (home; away) and the presence or absence of crowds. Post-hoc analyses reveal that referees awarded significantly more penalties to the away team compared to the home team when there is a crowd present. However, when there are no spectators, the number of penalties awarded to the away and home teams are not significantly different. In order to generalize these results, we took advantage of the extension of the pandemic and the unusual game setting it provided to observe the behavior of referees during the 2020–2021 regular season. Again, using data from the National Hockey League ($n = 1639$), but also expanding our sample to include Canadian Hockey League games ($n = 1709$), we also find that the advantage given to the home team by referees when in front of a crowd fades in the absence of spectators. These findings provide new evidence suggesting that social pressure does have an impact on referees' decision-making, thus contributing to explain the phenomenon of home advantage in professional ice hockey.

## Introduction

During the 2019 National Hockey League (NHL) playoffs, controversies surrounding the work of referees received a lot of scrutiny [1,2]. While blaming referees for a team's loss is a ritual as old as the game of hockey itself, several experts agree that the 2019 playoffs have been riddled with decisions that changed the outcome of many games [3,4]. With those controversies in mind, referees may have expected their work to be scrutinized even more closely than usual during the 2020 playoffs. Fortunately for them, the COVID-19 pandemic took some firsthand eyewitnesses out of the equation. Indeed, the 2020 playoffs were held in empty arenas, with no crowds allowed. While this may give referees a break from real-time criticism of their decisions

**Data Availability Statement:** The data underlying the results presented in the study are available from NHL official website (www.nhl.com) and from CHL (chl.ca) website subdivisions (whl.ca; ohl.ca; theqmjhl.ca).

**Funding:** This work was supported by the Natural Sciences and Engineering Research Council of Canada (NSERC) in the form of a grant awarded to DF (RGPIN-402513-2012), and by the Fonds de Recherche du Québec – Société et culture (FRQSC) in the form of a graduate scholarship awarded to JG (2021-B2Z-288052). The funders had no role in study design, data collection and analysis, decision to publish, or preparation of the manuscript.

**Competing interests:** The authors have declared that no competing interests exist.

by fans, it provides an unprecedent opportunity to examine the crowd's contribution to one of the most notorious phenomena in sport, the home advantage.

Home advantage is a well-documented concept in sports [5–8]. In hockey, playing at home provides advantages related to the rules. For example, players from the home team have the right to position themselves last when taking a face-off, a situation where the players attempt to gain control of the puck after it is dropped on the ice by the referee, which provides a technical upper-hand. In addition, home team coaches are allowed to make the last change of players after a stoppage of play, meaning they can choose to put on their desired players to match the opposing teams' choice of players. Essentially, these two rules allow the home team to observe the opposing team's strategy and adapt to it. In addition to rules-related benefits, playing in front of a local crowd is acknowledged to increase the probability of winning [9–11]. While the presence of fans can positively impact home team players effort-level [12], supporters can also have an impact on referee decisions [13–16]. The crowd's behaviour is processed by referees as social information, which in turn can influence their decision-making [17].

Since crowds are predominantly composed of local fans, their reactions are mainly in response to events that affect outcomes for the home team. Supporters conventionally cheer decisions benefitting the home team and boo decisions against them. Among the decisions that provoke the strongest reactions from crowds, are penalty calls. In hockey, penalties generate an important disadvantage for the offending team by offering a powerplay to the opposite team. In this situation, the penalized player must serve his penalty in the penalty box and his team cannot replace his presence on the ice. This means his team must perform with one less player for a period of time that is determined by the seriousness of the reprehensible act (2, 4 or 5 minutes). Crowd reactions to what fans consider illegal maneuvers may act as salient cues for referees, having the potential to influence their decisions on penalty calls [16].

Referees' decisions leading to penalty calls, which then result in the home team benefitting from more powerplay opportunities than the visiting team, are considered a contributing factor to home advantage in the NHL [12,14,18]. A plausible justification for that observation is the social pressure exerted by the home crowd. The presence of supporters in sporting events increase social pressure on referees, a phenomenon reputed to influence decisions in favor of the home team [19–23]. This pressure could lead referees to internalize crowd preferences in their decision-making [24]. Since calling penalties is a decision that relies entirely on the judgment of a referee, analyzing this type of decision offers an opportunity to assess how the crowd may influence NHL referees' decision-making processes.

Although the effect of social pressure exerted by crowds on NHL referees' behavior has already been investigated [14,18], completely isolating the impact of supporters is methodologically challenging. While controlling for different characteristics of the crowd is possible, totally removing supporters from NHL arenas is almost unimaginable. That is until the COVID-19 pandemic became problematic, and the implementation of social distancing measures forced the NHL to rethink the way their 2020 playoffs games were to be played. The NHL designated the cities of Edmonton and Toronto in Canada as hub cities. These hub cities allowed the NHL to control social distancing by holding the playoffs in empty arenas, creating an ecological environment where the impact of the home crowd was completely removed. This context offered a unique opportunity to analyze how referees make decisions in a context without direct social pressure from fans.

The purpose of this study is to investigate whether the presence of a crowd can influence the decision-making of NHL referees. This is done by taking advantage of the 2020 playoffs being played in front of empty seats, and comparing the number of penalties called by NHL referees against the home and away teams respectively in 2020 to the previous five years (2015 to 2019). Based on previous studies, we hypothesize that playing in crowded arenas leads to a

reduction in penalties called against the home team and that this trend will disappear in a context where no local fans are present to exert social pressure on the referees. This would validate to a certain extend that the presence of crowds at professional ice hockey games do have an impact on the referees' decision-making, thus reinforcing the concept of home advantage in professional sports. To ensure the generalizability of the results, analysis of referee behavior in various contexts is also presented. This was done by comparing 2020–2021 regular season games from the NHL, as well as from the Canadian Hockey League (CHL), the top junior ice hockey league in Canada, to games from their respective previous regular seasons. Those additional data sets offer the possibility to observe penalties awarded by referees in different contexts, when compared to the initial sample. Notably, the new data sets looked at games in which the stakes are minimized compared to the playoffs, at games played in the teams' local arenas rather than in neutral hub cities, and at games where the playing level and the degree of expertise of the referees are lower.

## Methods and results

### NHL playoffs data set

Our database includes all NHL playoff games since 2015, giving five years of data prior to the COVID-19 pandemic. As suggested by Lakens and Evers [25], the 547 games played during this 6-year period are sufficient to achieve 80% power to observe a small effect size with an alpha of .05. This relatively short period also allows for consistency in the referees assigned to playoff games. We accessed NHL official web site (nhl.com) to obtain game reports from 2015 to 2019 games, played in front of a live audience, and from 2020 games, played without crowds.

Due to the shortened season, the 2020 playoffs were held under an adjusted format. First, home and away status was granted as if the games were played in the normal playoffs format. This means that even though teams were not playing in their home cities, the higher ranked team was given home status for games 1, 2, 5 and 7 of the best-of-7 series, while the lower ranked team was given home status for games 3, 4 and 6.

Second, the number of eligible teams was increased, and the number of games was slightly modified. Eight teams were added to the 16 teams normally qualified for the playoffs. Of the 24 teams that qualified in 2020, the top 4 of the eastern conference and the top 4 of the western conference got a free pass to the first round. However, those teams had to play a round-robin, where each team plays the other top three teams in their conference, to determine their final ranking for the rest of the playoffs. Those twelve round-robin games, 6 per conference, have not been considered in our data set since these games did not have the same stakes as a standard series where the losing team is eliminated. As the importance of these games is diminished, the behavior of the referees could be different compared to series where the stakes are higher. In addition, these games were played under regular season rules when it came to overtime. More specifically, instead of playing 5-on-5 until a team scores a goal like in other playoff games, round-robin overtime periods were played 3-on-3 for a maximum of 5 minutes, followed by a shootout if the tie persists. This rule restricts the number of minutes played in the event of a tie compared to normal playoffs, thus limiting the opportunity for referees to award penalties. Nevertheless, we conducted additional statistical analyses, which included the twelve round-robin games, confirming that their exclusion did not affect the pattern of findings.

The remaining 16 teams, representing the bottom eight teams in each conference, competed in a qualifying round consisting of best-of-five series, with the winners continuing on to the first round. Because best-of-five games lead to a team's elimination and have the same importance and rules as normal playoff games, we included these games in the data set. We

**Table 1. Penalty kills per year during the NHL playoffs.**

| Year | Games | Crowd | Away PK | Home PK | Mean Away PK | Mean Home PK |
|------|-------|-------|---------|---------|--------------|--------------|
| 2015 | 89 | Yes | 276 | 244 | 3.10 (1.438) | 2.74 (1.434) |
| 2016 | 91 | Yes | 313 | 283 | 3.44 (1.551) | 3.11 (1.567) |
| 2017 | 87 | Yes | 296 | 250 | 3.40 (1.521) | 2.87 (1.310) |
| 2018 | 84 | Yes | 291 | 252 | 3.46 (1.766) | 3.00 (1.575) |
| 2019 | 87 | Yes | 306 | 243 | 3.52 (1.704) | 2.79 (1.511) |
| 2020 | 109 | No | 380 | 389 | 3.49 (1.772) | 3.57 (1.612) |

Standard deviations are presented in parentheses.

excluded games involving the Edmonton Oilers and Toronto Maple Leafs from the data set, as those teams were the only ones playing in their home arenas. As with round-robin matches, excluding this data from statistical analyses did not change the results.

All penalties generating powerplays were collected from NHL official game summaries. We converted each game's powerplay opportunities for one team into the number of penalties taken by the opposing team. This ensured that only penalties creating a numerical disadvantage for the offending team were retained. These are also known as penalty kills (PK). We rejected all penalties where players on both teams were being punished and neither team had a power play, specifically minor double penalties and fights. We also rejected 10-minute penalties for misconduct since these types of penalties do not generate powerplay opportunities for the opposing team. Descriptive statistics per year are reported in Table 1.

Our data also includes dummy variables for each team in each year, to control the characteristics of the teams that may contribute to penalty decisions by NHL referees. For example, previous results suggest that the relative strength of teams may contribute to differences in playing style, resulting in more penalties being awarded to the weaker team [26]. Style of play not related to a team's relative strength, but rather related to the offensive system of play favored by the team, has also been identified as a key factor in home advantage in the National Basketball Association [27], suggesting that the same effect could be observed in the NHL. Finally, dummy variables for game referees have been included to serve as fixed effects to control for individual propensity to call penalties. Individual differences exist in the tendency of referees to call more penalties to home or away teams [26].

## NHL playoffs model

To estimate the number of penalties called by referees, we used a Poisson generalized linear mixed model (GLMM) with a log link function. The model provides positive fitted values through the logarithmic link, and the Poisson distribution best fits the count data in a fixed period of time, in this case the number of penalties awarded in an NHL game [28,29]. Standard Poisson regression was selected, since the model meets assumption of equidispersion and data were not zero inflated or zero truncated [28,29]. Our model is represented by:

$$\ln(\mu_{i,j}) = \text{Home}_{i,j} + \text{Crowd}_j + \text{Home}_{i,j} * \text{Crowd}_j + \text{Team}_k + \text{Ref1}_j + \text{Ref2}_j + \epsilon_{i,j,k} \qquad (1)$$

where $\mu_{i,j}$ indicates the penalties awarded that resulted in penalty kills. Independent categorical variables are $Home_{i,j}$, representing the status of the team $i$ (home = 0; away = 1), and $Crowd_j$ indicating the presence or absence of an audience in match $j$ (spectators = 0; no spectator = 1), with each match being counted twice, once for the away team and once for the home team [24,30]. The interaction between these variables is also included in the model to capture the

effect on referee decisions of the status of teams, with and without crowd. $Team_k$ fixed effects for each team during year $k$ are included to control for style of play. Fixed effects for both referees were also included to control for differences in propensity to award penalties, with $Ref1_j$ and $Ref2_j$. We use the package *stats* included in the software R to fit the GLMM [31], the package *rsq* to estimate the adjusted $R^2$ [32], and the package *emmeans* [33] to run post-hoc analyses.

## NHL playoffs analysis and results

The model revealed a significant effect of the *Home*Crowd* interaction (b = .17, *p* = .024) for NHL playoff games. The results suggest that the likelihood of penalties being awarded by NHL referees differs depending on the status of the team and the presence or absence of spectators. Results of the NHL playoffs Poisson GLMM are presented in Table 2.

Simultaneous pairwise comparisons using Tukey's HSD test indicated that in front of a crowd, penalties awarded to the away team were significantly higher than those awarded to the home team (Z = 4.022, *p* < .001). In contrast, no significant difference is observed when teams played in absence of the crowd (Z = -.434, *p* = .973). Referees also award significantly fewer penalties to the home team when they are supported by the crowd, compared to the home team playing in front of empty seats (Z = -2.833, *p* = .024). A similar trend can be observed when comparing the home team supported by a crowd with the opposing team playing in an empty arena (Z = 2.462, *p* = .066). However, there is no difference in the penalties awarded to the visiting team when playing in front of a crowd compared to an empty arena (Z = -.475, *p* = .965), nor in comparison to the home team playing without spectators (Z = -.877, *p* = .817). The model explains 11.3% of the variance in penalties called by NHL referees, which is considered a small effect size [34]. A small effect size was to be expected, as an unbiased referee should only consider the actions of the players and eliminate all external factors from his environment, i.e., the variables in the model. The main explanation for penalty calls by a referee remains the actual presence of illegal actions by the players, a variable that cannot be included in the model, as no objective data exists to that effect.

Fig 1 presents a violin plot with mean penalty kills (±1 SD error bars) for away and home teams, in front of crowds before the pandemic and in empty arenas during the 2020 playoffs.

Our model adds evidence that referee decision-making plays a part in explaining the concept of home advantage by showing that in front of local supporters during the playoffs, NHL referees tend to award less penalties to the home team compared to the away team, as well as

**Table 2. Estimated regression parameters for NHL playoffs Poisson GLMM (*n* = 1094).**

| | Estimate (b) | Std. error | *z* value | *P*-value | IRR Exp(b) |
|---|---|---|---|---|---|
| Intercept | 1.388 | 0.196 | 7.089 | <0.001 | 4.006 |
| Home | -0.155 | 0.038 | -4.022 | <0.001 | 0.857 |
| Crowd | 0.077 | 0.269 | 0.285 | 0.775 | 1.080 |
| Home: Crowd | 0.186 | 0.083 | 2.254 | 0.024 | 1.205 |
| Referees FE | Yes | | | | |
| Team FE | Yes | | | | |
| Adjusted $R^2$ | 0.113 | | | | |

Estimates are in log-odds from a Poisson regression.

FE: Fixed effect.

IRR: Incidence rate ratio.

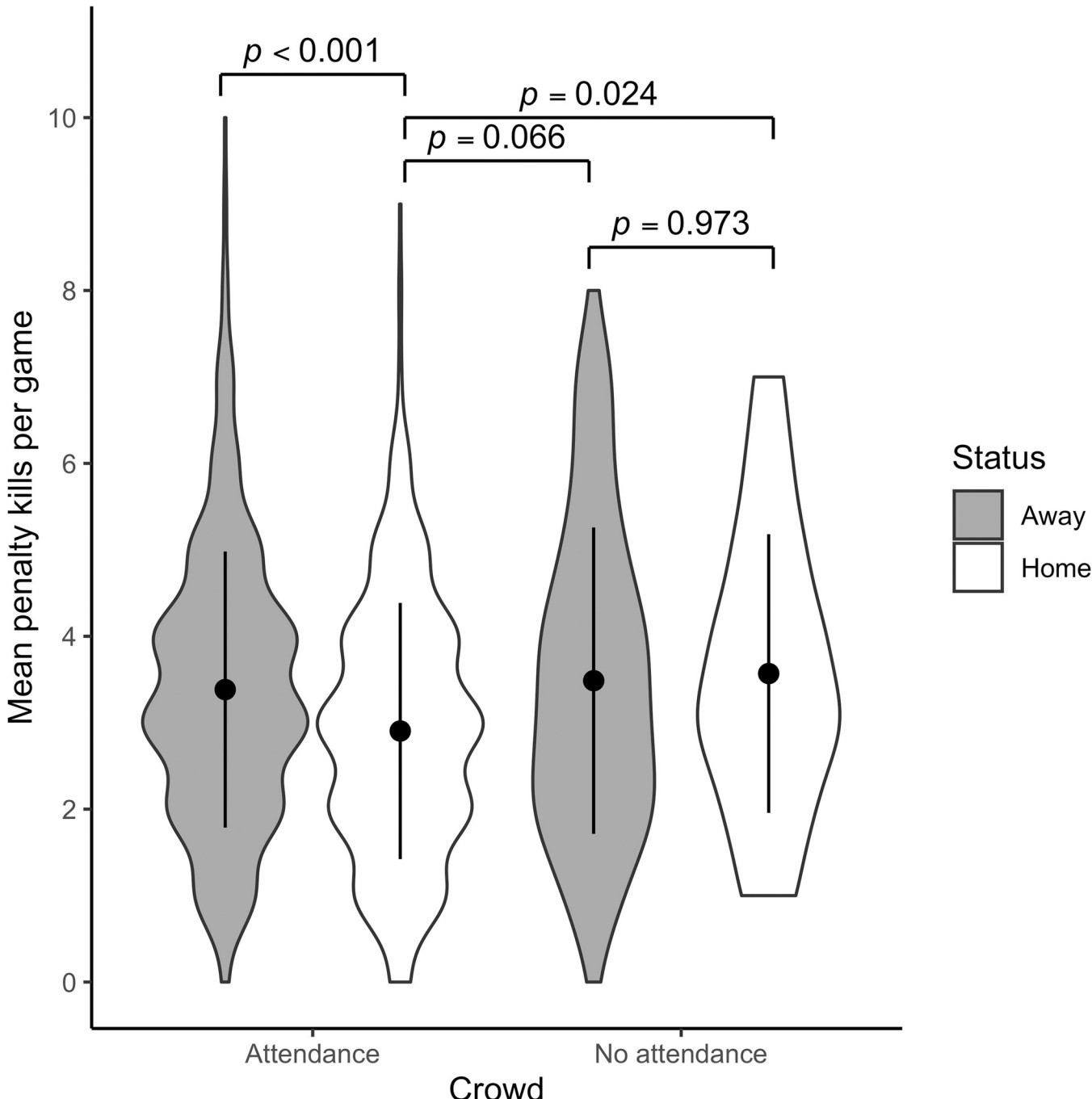

**Fig 1. Mean penalty kills per team with and without attendance during the NHL playoffs.** For the sake of simplicity, the raw data have been presented in the figure, even though the model results in Tables 2 and 3 are in log-odds.

both teams when the seats are empty. The results also support our hypothesis, which suggests that the home team is no more privileged when it comes to penalty calls when there is no home crowd in the building. Our model offers a good way to define the role of referees by controlling for the style of play of the teams, and the individual propensity of referees to award penalties.

**Table 3. Post-hoc results of the NHL model during playoffs ($n$ = 1094).**

| | Estimate (b) | Std. error | z ratio | P-value | IRR Exp(b) |
|---|---|---|---|---|---|
| Away Crowd–Away Empty | -0.037 | 0.077 | -0.475 | 0.965 | 0.964 |
| Away Crowd–Home Crowd | 0.155 | 0.039 | 4.022 | <0.001 | 1.168 |
| Away Crowd–Home Empty | -0.068 | 0.078 | -0.877 | 0.817 | 0.934 |
| Away Empty–Home Crowd | 0.191 | 0.078 | 2.462 | 0.066 | 1.210 |
| Away Empty–Home Empty | -0.032 | 0.073 | -0.434 | 0.973 | 0.969 |
| Home Crowd–Home Empty | -0.222 | 0.079 | -2.833 | 0.024 | 0.801 |

Results are averaged over the levels of: *Ref1*, *Ref2*, *Team*.

Estimates are in log-odds.

## Regular season data sets

To investigate the generalizability of our results, we used the same strategy and variables as for the NHL playoff data set to construct additional data sets for the NHL and CHL regular seasons. First, we aimed to see if crowd pressure had an impact on referee behavior in games where the stakes are lower and where games are played in the local teams' arenas rather than in neutral hub cities. Secondly, we wanted to test whether the influence of spectators was observable on referee decision-making in a league below the NHL level. The collection of data from games played in empty arenas was made possible by the extension of restrictive health measures related to the COVID-19 pandemic that forced some hockey leagues to begin the 2020–2021 season without crowds. This is the case for the NHL and CHL. In addition to having to play some games in front of empty seats, these leagues have also been compelled to shorten their regular schedule to avoid cancelling their seasons altogether. For the NHL, we once again used the official website nhl.com to collect the data. As for the Canadian Hockey League, we used the subdivisions of the official website (chl.ca) to acquire game summaries from the individual leagues that make up the CHL (whl.ca, ohl.ca, theqmjhl.ca).

We started by collecting data for the 2020–2021 NHL season, in which teams were schedule to play 56 games instead of the usual 82 games that are normally played in non-pandemic regular season. Unlike the playoffs, no hub city was designated, and teams played in their respective hometown. This is closer to the usual format of NHL games, providing an opportunity to verify that the 2020 playoff results are not due to hub cities being neutral ground. The NHL's four divisions were reformatted with the goal of having teams exclusively play against other teams in their division. Given that government restrictions prohibited border crossings between the Canada and the United States, an exclusively Canadian division was created, along with three American divisions. All the games in the Canadian division were played in empty arenas. For the American division games, state-specific rules dictated whether or not spectators were present, with some games played in front of limited crowds ($n$ = 310). These games were removed from the sample because the very limited number of fans might not generate the same pressure on the referees as in an arena full of spectators. Nevertheless, additional statistical analyses that include these games were performed to confirm that removing them does not change the results.

Next, we compiled the data for the 2020–2021 CHL season. The CHL is made up of three leagues spread across Canada and the United States. The Western Hockey League (WHL), the Ontario Hockey League (OHL) and the Quebec Major Junior Hockey League (QMJHL) includes teams from nine provinces of Canada and four American states. As public health measures vary between provinces and states, the structure of the regular schedule also varied between the 3 leagues that make up the CHL. For starters, health measures caused the OHL to

completely cancel its 2020–2021 season. For their part, the WHL and QJMHL opted for a shorter schedule. Since the teams that make up these two leagues are spread across several Canadian provinces and some American states, the format of the games varied according to the public health measures of each province and state. As a result, some of the games in the QJMHL were played in front of fans ($n$ = 102). These games were removed from the sample. Once again, additional statistical analysis including the data from these games confirms that their removal does not alter the significance level of the results. Public health restrictions also led both leagues to hold some games in hub cities, similar to the NHL during its 2020 playoffs, while the rest of the games were held in the respective teams' hometowns.

In order to compare decisions taken by referees in this unusual context with decisions taken by referees during typical games played in front of crowds, we used the entirety of the previous season's games (2019–2020 regular season) for both the NHL and CHL. For the CHL, we excluded OHL games since it did not play games in 2020–2021 and meant it could not be used for comparison purposes. Although the 2019–2020 seasons were somewhat shortened due to the outbreak of the pandemic, the number of games in each league is far greater than the number of games suggested by Lakens and Evers [25] to observe a small effect size. Descriptive statistics for the regular seasons are presented in Table 4.

## Regular seasons model

The same Poisson GLMM, represented by Eq (1), was used to estimate the number of penalties called by NHL and CHL referees during regular seasons. The model for both data sets still meets assumption of equidispersion and data were not zero inflated or zero truncated [28,29]. The R packages *rsq* [32] and *emmeans* [33] were again used to estimate the adjusted $R^2$ and to run post-hoc analyses, respectively.

## Regular seasons analysis and results

The model illustrates a significant effect of the *Home*Crowd* interaction for NHL regular season games (b = .10, $p$ = .010) and for CHL regular season games (b = .10, $p$ = .018). The results suggest that the probability of penalties being called by referees in these two leagues is modified by team status and the presence or absence of fans. The Poisson GLMM results for both leagues in regular season are presented in Table 5.

Simultaneous pairwise comparisons using Tukey's HSD test indicated that in front of a crowd, penalties awarded to the away team were significantly higher than those awarded to the home team in both the NHL (Z = 3.729, $p$ = .001) and CHL (Z = 3.251, $p$ = .006). Similarly to the 2020 NHL playoffs, these differences fade in regular season games where arenas are empty in both the NHL (Z = -.230, $p$ = .996) and the CHL (Z = -.820, $p$ = .845). In contrast to the

**Table 4. Penalty kills per year during the NHL and CHL regular seasons.**

| Year | Games | Crowd | Away PK | Home PK | Mean Away PK | Mean Home PK |
|---|---|---|---|---|---|---|
| **NHL** | | | | | | |
| 2019–2020 | 1082 | Yes | 3368 | 3064 | 3.11 (1.484) | 2.83 (1.353) |
| 2020–2021 | 557 | No | 1634 | 1643 | 2.93 (1.484) | 2.95 (1.452) |
| **CHL** | | | | | | |
| 2019–2020 | 1264 | Yes | 4979 | 4651 | 3.94 (1.791) | 3.68 (1.648) |
| 2020–2021 | 445 | No | 1660 | 1713 | 3.73 (1.727) | 3.85 (1.830) |

Standard deviations are presented in parentheses.

**Table 5. Estimated regression parameters for NHL (*n* = 3278) and CHL (*n* = 3418) regular seasons Poisson GLMM.**

| | Estimate (b) | Std. error | | *z* value | *P*-value | IRR Exp(b) |
|---|---|---|---|---|---|---|
| **NHL** | | | | | | |
| Intercept | 1.033 | 0.111 | | 9.437 | <0.001 | 2.851 |
| Home | -0.093 | 0.022 | | -4.179 | <0.001 | 0.911 |
| Crowd | -0.068 | 0.040 | | -0.114 | 0.909 | 0.995 |
| Home: Crowd | 0.101 | 0.043 | | 2.571 | 0.010 | 1.116 |
| Referees FE | | | Yes | | | |
| Team FE | | | Yes | | | |
| Adjusted R$^2$ | | | 0.081 | | | |
| **CHL** | | | | | | |
| Intercept | 0.320 | 0.580 | | 0.552 | 0.581 | 1.377 |
| Home | -0.066 | 0.020 | | -3.251 | 0.001 | 0.936 |
| Crowd | -0.035 | 0.034 | | -1.030 | 0.303 | 0.966 |
| Home: Crowd | 0.095 | 0.040 | | 2.362 | 0.018 | 1.099 |
| Referees FE | | | Yes | | | |
| Team FE | | | Yes | | | |
| Adjusted R$^2$ | | | 0.077 | | | |

Estimates are in log-odds from a Poisson regression.

FE: Fixed effect.

IRR: Incidence rate ratio.

NHL playoffs, no other significant differences are observed in the regular season in either the NHL or the CHL. The model for NHL explains 8.1% of the variance, while the model for CHL measures 7.7% of the variance. Again, a small effect size is observable, reinforcing the idea that external factors generate only a small influence on referee penalty calls.

Figs 2 and 3 present violin plots with mean penalty kills (±1 SD error bars) for away and home teams, in front of crowds before the pandemic and in empty arenas during the 2020–2021 regular seasons.

Consistent with the results stemming from the 2020 NHL playoffs, NHL and CHL referees, during regular season games, award the home team significantly fewer penalties on average compared to the away team. The results again corroborate our hypothesis that the home team is no longer favored when it comes to penalty calls in situations where there are no local fans in the arena.

## Discussion

Home advantage is a well-established concept in professional sport and the influence of the home crowd on referee decision-making is a factor that is assumed to contribute to this phenomenon. Using a unique ecological context generated by the COVID-19 pandemic, we found evidence of the effect of crowds on NHL referees' decisions during the playoffs. Our observations can also be generalized to different contexts, i.e., regular season and lower-level league. These findings add to existing scientific evidence supporting the involvement of sports referees in home advantage [30,35–39] and reinforce the existence of this phenomenon in elite ice hockey. The results also allow us to better understand the decision-making processes of ice hockey referees, who have the potential to change the outcome of games.

Our findings provide new evidence linking referees' decision-making with the concept of home advantage in elite ice hockey. More precisely, our model suggests that the home team

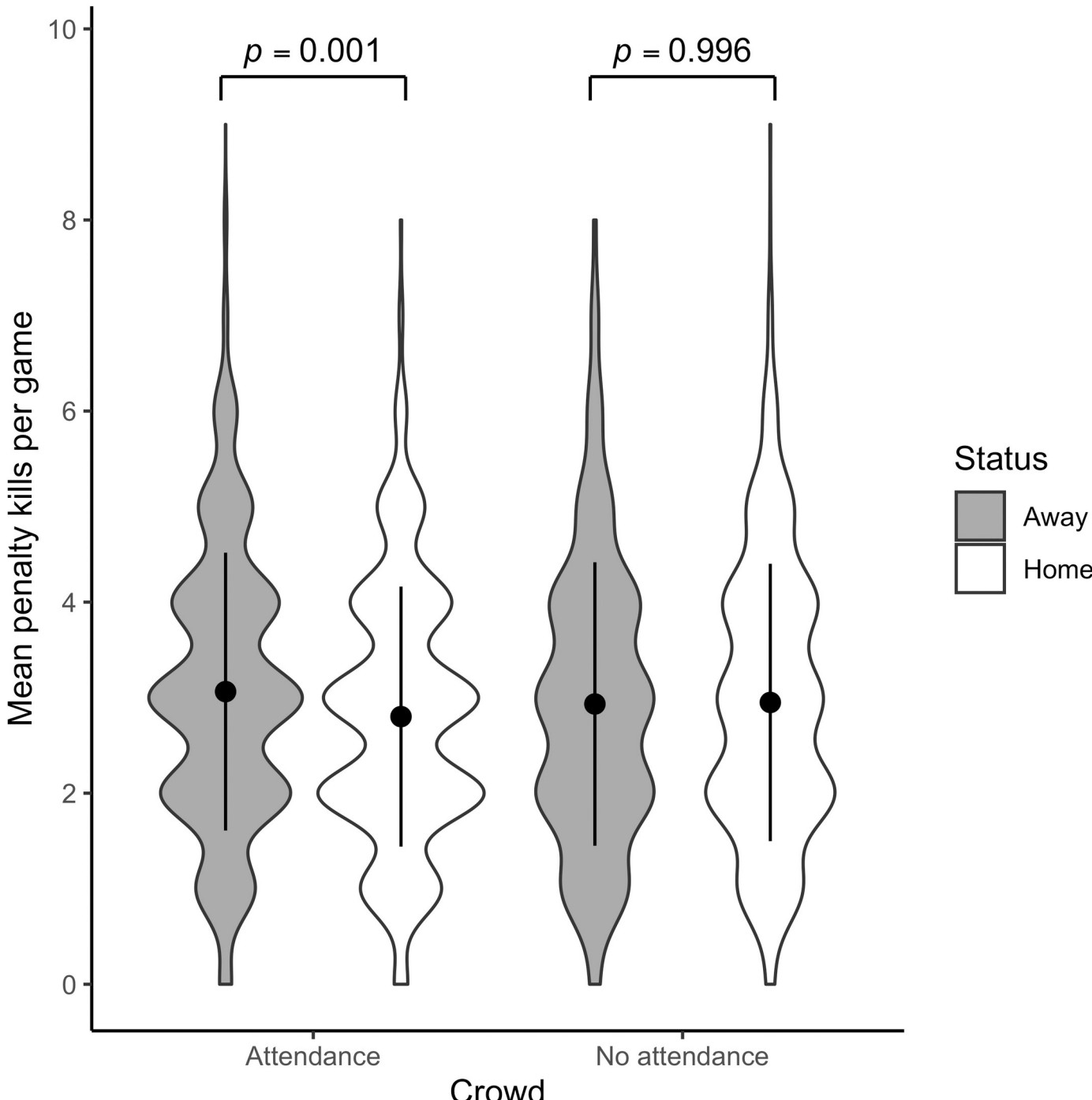

**Fig 2. Mean penalty kills per team with and without attendance during the NHL regular season.** For the sake of simplicity, the raw data have been presented in the figure, even though the model results in Tables 5 and 6 are in log-odds.

receives less penalties when they play in front of their home crowd compared to the opposing team. This advantage disappears when the games are played without spectators. These results suggest that referees are more lenient towards home players when they are playing in front of their local fans.

**Table 6. Post-hoc results of the NHL (*n* = 3278) and CHL (*n* = 3418) models during regular seasons.**

| | Estimate (b) | Std. error | *z* ratio | *P*-value | IRR Exp(b) |
|---|---|---|---|---|---|
| **NHL** | | | | | |
| Away Crowd–Away Empty | 0.068 | 0.031 | 2.181 | 0.129 | 1.070 |
| Away Crowd–Home Crowd | 0.093 | 0.025 | 3.729 | 0.001 | 1.098 |
| Away Crowd–Home Empty | 0.060 | 0.032 | 1.899 | 0.228 | 1.062 |
| Away Empty–Home Crowd | 0.025 | 0.032 | 0.790 | 0.859 | 1.025 |
| Away Empty–Home Empty | -0.008 | 0.035 | -0.230 | 0.996 | 0.992 |
| Home Crowd–Home Empty | -0.033 | 0.032 | -1.037 | 0.728 | 0.967 |
| **CHL** | | | | | |
| Away Crowd–Away Empty | 0.035 | 0.034 | 1.030 | 0.732 | 1.036 |
| Away Crowd–Home Crowd | 0.066 | 0.020 | 3.251 | 0.006 | 1.069 |
| Away Crowd–Home Empty | 0.007 | 0.034 | 0.199 | 0.997 | 1.007 |
| Away Empty–Home Crowd | 0.031 | 0.034 | 0.918 | 0.795 | 1.032 |
| Away Empty–Home Empty | -0.028 | 0.035 | -0.820 | 0.845 | 0.972 |
| Home Crowd–Home Empty | -0.060 | 0.034 | -1.755 | 0.236 | 0.942 |

Results are averaged over the levels of: *Ref1*, *Ref2*, *Team*.

Estimates are in log-odds.

The social pressure exerted by fans on referees is a plausible explanation for their behavior [19–23]. Crowd pressure seems to have an impact on referees in different ice hockey leagues, as well as in competitive settings with varying stakes. However, the behavior of referees seems to be affected differently depending on the context. In the regular season, despite the difference in penalties called between the home and visiting teams when the crowd is present, it is difficult to determine whether the referees are calling more penalties on the visiting team, fewer penalties on the home team, or a combination of both. During the playoffs, a more specific direction of bias towards the home team is identifiable. Consistent with our results, crowd pressure during the playoffs can be observed by a decrease in the number of penalties awarded by the referees to the home team, a phenomenon that can be identified as an omission bias [40]. This bias would be more prevalent in critical moments, such as playoffs, where referees are known to try to limit their impact on the flow of the game [41]. Moreover, referees who display omission biases might be recognized as being fairer than referees who exhibit other types of biases. Indeed, the omission of penalties imposed on the home team can be considered less intrusive or damaging to the visiting team than an act of commission, such as an increase in the penalties imposed on the latter [41]. An excellent example of the perceived immorality of a commission call is the 2021 dismissal of Tim Peel, an NHL referee, who was caught admitting that he was trying to balance out the number of penalties awarded during the game by giving a questionable penalty to one of the teams [42]. To this day, there has never been a dismissal of an NHL referee who willingly omitted to call a penalty, or as they say in sports jargon, "swallowed his whistle" [41]. However, these examples raise the need to be cautious in interpreting an omission bias among NHL referees during the playoffs. While, as mentioned above, the greater stakes of playoff games may be sufficient to explain the omission bias, it is also important to consider that the referees selected to officiate these games are those whom the league considers their best. These referees can be perceived as being the best precisely because they tend to omit penalties to the home team, rather than awarding more penalties to the opposing team, which is considered more morally acceptable. In this case, the presence of the omission bias during the playoffs could be due to the selection of referees by NHL

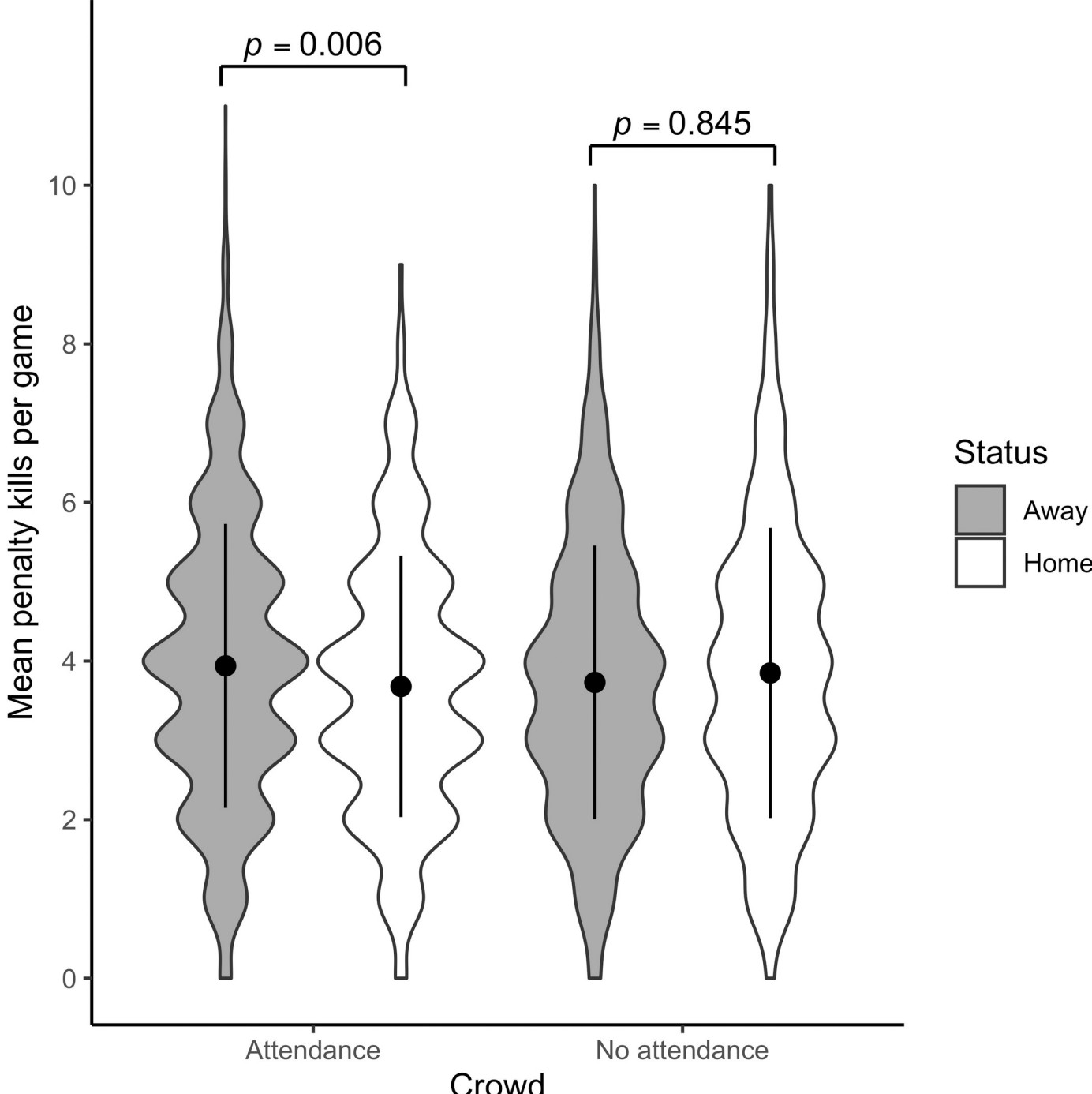

**Fig 3. Mean penalty kills per team with and without attendance during the CHL regular season.** For the sake of simplicity, the raw data have been presented in the figure, even though the model results in Tables 5 and 6 are in log-odds.

executives, rather than the higher stakes of the games being officiated. It would be relevant to assess this in future studies.

Given that the results are based on a unique ecological environment, the number of games played without spectators remains low compared to the data available for usual games played

in front of a crowd. Also, since playoff games played without an audience are limited to the NHL in 2020, our findings in playoffs are not generalizable beyond a reasonable doubt to other ice hockey leagues. As far as we know, the NHL is the only league to have played playoffs in 2020. In 2021, the professional leagues that have held playoffs have done so with a small number of games or in front of a limited number of fans. In the CHL, only the QMJHL has held its playoffs in an adjusted format, limiting the number of games in empty arenas to 44, which does not provide enough statistical power for analysis. Also, our sample is limited to NHL and CHL referees, meaning that other referees in minor and major hockey leagues may behave differently. Furthermore, given that these leagues are played in North America, referees may be influenced differently than those in other continents. Since the style of refereeing may be influenced by cultural factors, it could be conceivable to expect different results from European referees, for example. Our project is limited to the NHL and CHL, and more research needs to be conducted in different leagues and countries before we can generalize our results to the broad population of ice hockey referees.

However, our study has the obvious strength of presenting an ecological design made possible by the COVID-19 pandemic. For the very first time in studies relating to elite referee behavior, the local crowd and its possible impact on decision-making was physically eliminated from the equation. This offers a significant advantage over designs where audience influence had to be statistically controlled. The statistical model used also adds strength to our results, whereas previous studies supporting the influence of the public on referees' decision-making, mainly those made in soccer, have relied on ordinary linear regression models that does not fit the count data as well as Poisson regression [28]. Moreover, our study examined the behavior of referees working for the world's best professional ice hockey league, as well as for one of the top junior hockey leagues. These referees are considered to be amongst the best in their discipline and the least likely to be influenced by factors outside of the game itself. The fact that our results present an impact of the crowd on elite referees' decision-making suggests that this impact should also be present, and possibly greater, in less experienced or capable referees. Finally, our study exposes the effect of crowds on referee decisions in a variety of contexts. Social pressure from the crowd influences referees in more than one league, in leagues of different levels, as well as in games with different stakes. Our research project illustrates that this impact disappears with the absence of crowds, both in neutral hub cities and in hometown arenas.

## Conclusion

Home advantage can be explained in part by the social pressure exerted by home crowds on referees. A more exhaustive knowledge of which decisions are more predisposed to be influenced by the presence of crowds could help to better train referees, enabling them to deal with the existence of decision biases. With the advancement of technology, the work of referees is now highly scrutinized and improving their ability to make impartial decisions on every call can have a positive impact on both the length of their careers and the reputation of the league itself.

## Acknowledgments

We would like to thank our collaborators, Jason Lacelle and Francis Gingras, for reviewing the text and providing constructive comments.

## Author Contributions

**Conceptualization:** Joël Guérette, Caroline Blais, Daniel Fiset.

**Data curation:** Joël Guérette.

**Formal analysis:** Joël Guérette, Caroline Blais, Daniel Fiset.

**Investigation:** Joël Guérette.

**Methodology:** Joël Guérette, Caroline Blais, Daniel Fiset.

**Project administration:** Joël Guérette, Caroline Blais, Daniel Fiset.

**Resources:** Joël Guérette.

**Software:** Joël Guérette.

**Supervision:** Caroline Blais, Daniel Fiset.

**Validation:** Caroline Blais, Daniel Fiset.

**Visualization:** Joël Guérette.

**Writing – original draft:** Joël Guérette.

**Writing – review & editing:** Joël Guérette, Caroline Blais, Daniel Fiset.

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
