## [Decision Letter · Decision Letter 0]

10 Jun 2021

PONE-D-21-15462

The absence of fans removes the home advantage associated with penalties called by National Hockey League referees

PLOS ONE

Dear Dr. Fiset,

Thank you for submitting your manuscript to PLOS ONE. After careful consideration, we feel that it has merit but does not fully meet PLOS ONE’s publication criteria as it currently stands. Therefore, we invite you to submit a revised version of the manuscript that addresses the points raised during the review process.

Your paper is interesting, clear and well written. However, we recommend to address the concerns from the reviewers. For example, to apply the proposed methodology to other datasets, to demonstrate the impact of no crowd allowance in more sports or leagues.

We look forward to receiving your revised manuscript.

Kind regards,

Luca Pappalardo

Academic Editor

PLOS ONE

Journal Requirements:

"We would also like to thank NSERC and FRQSC for their financial support."

"This work was supported by the Natural Sciences and Engineering Research Council of Canada (NSERC) in the form of a grant awarded to DF (RGPIN-402513-2012), and by the Fonds de Recherche du Québec – Société et culture (FRQSC) in the form of a graduate scholarship awarded to JG. The funders had no role in study design, data collection and analysis, decision to publish, or preparation of the manuscript."

Additional Editor Comments (if provided):

All reviewers find the paper interesting, clear and well written. However, they also ask some improvements. In particular, one of the reviewers asks to apply the proposed methodology to other datasets and sports, to demonstrate the impact of no crowd allowance in more sports or leagues.

Reviewers' comments:

Reviewer's Responses to Questions

**Comments to the Author**

1. Is the manuscript technically sound, and do the data support the conclusions?

Reviewer #1: Yes

Reviewer #2: Yes

2. Has the statistical analysis been performed appropriately and rigorously? 

Reviewer #1: Yes

Reviewer #2: Yes

3. Have the authors made all data underlying the findings in their manuscript fully available?

Reviewer #1: Yes

Reviewer #2: No

4. Is the manuscript presented in an intelligible fashion and written in standard English?

Reviewer #1: Yes

Reviewer #2: Yes

5. Review Comments to the Author

Reviewer #1: The paper is well written and easy to follow. Only a few change should be made:

- Lines 156-175. This paragraph describes the model fitted. In this part, no results are presented. I suggest moving it in the method section.

- Line 183-201. Please provide a table of the post-hoc analysis. You could also add statistical difference references in the Figure 1.

- Line 195. The model proposed explain only the 11.3% of the total variance. Is it statistically significant?

Reviewer #2: In this paper authors address a valuable issue in team sports, i.e. pressure exerted from crowd to referees. In this case, authors focus on Ice Hockey. In particular, they analyze the impact of COVID-19 restrictions regarding to game attendance during NHL play-offs. The hypothesis that has been proven is realted to the different distribution of penalties awarded to home and away teams among crowded and non-crowded games. Authors show how the home advantage on that particular game event is not present during non-crowded games, thus highlighting the impact of crowd pressure on referees decisions.

The paper is well written and a pleasure to read. The method section is clearly explainend and the conclusions are sound. The use of a Poisson Generalized Linear Mixed Model to estimate the impact of every variable on referess decisions is properly motivated. However, methods such as SHAP would be interesting to test for this application.

In my opinion, to be mature for publication this paper should cover a broader area, highlighting the impact of no crowd allowance in more sports or leagues.

6. PLOS authors have the option to publish the peer review history of their article (what does this mean?). If published, this will include your full peer review and any attached files.

Reviewer #1: No

Reviewer #2: No

---

## [Author Response · Author response to Decision Letter 0]

23 Jul 2021

Reviewer #1

● The reviewer suggests moving lines 156-175 to the method section, since the paragraph describes the model and does not present results.

We fully agree with the reviewer that the paragraph in question is better positioned in the method section. Following reviewer #2's comments, we have modified our headings and now have a section entitled "Methods and Results". In this section, we have created a subheading entitled "NHL playoffs model" where lines 156-175 of the first manuscript now appear. This section can be seen on pages 8-9.

● Reviewer #1 requests a table for post-hoc results and suggests adding statistical difference references to Figure 1.

We thank reviewer #1 for their helpful suggestions. Both changes have been applied and can be seen on pages 10-11.

● The reviewer asks if the variance explained by the model is statistically significant at only 11.3%.

We agree that the variance explained by the model is low and understand the reviewer's questioning. However, this was expected, considering that factors external to the action of the hockey game (here, mainly the home/away status of the team and the presence/absence of the crowd) should not have an impact on the referees' decisions. The element that should explain the most variance in penalty calls is represented by the illegal actions of the players. Every penalty called by an unbiased referee should be earned, while every time an unbiased referee does not call a penalty, it should be because no illegal action was committed. Our model does not allow us to judge whether the penalties awarded are deserved or not, since no objective data are available for this purpose. 

It is also important to consider it is most likely when ambiguous situations arise that referees tend to be influenced by external factors. In our model, all penalties are included, even those that leave little room for interpretation by the referees (e.g., high stick penalties, penalties for delaying the game by directing the puck into the stands, etc.). Even considering these non-ambiguous penalties, our model still explains 11.3% of the variance. We therefore think that the external factors included in our model, i.e., the status of the team (home/away) and the presence or absence of the crowd, significantly influence the referees' decision-making.

Since the results raised questions from Reviewer #1, we decided to improve the section on variance in the text. Our modifications appear on lines 219-223 and can be read as follows:

“A small effect size was to be expected, as an unbiased referee should only consider the actions of the players and eliminate all external factors from his environment, i.e., the variables in the model. The main explanation for penalty calls by a referee remains the actual presence of illegal actions by the players, a variable that cannot be included in the model, as no objective data exists to that effect.”

Reviewer #2

● Reviewer #2 considers that the choice of the Poisson Generalized Linear Mixed Model is properly motivated. However, the reviewer mentions that other methods, such as SHAP, could have been interesting to test.

We thank reviewer #2 for this suggestion. We consider that other methods could indeed have been used to address our research question and believe that the SHAP method is an excellent recommendation. Since our method is considered adequate by Reviewer #2, we have stuck to the Poisson model for this paper. However, we will consider using other methods, such as SHAP, in future projects and invite other researchers to pursue our approach with other methods.

● Reviewer #2 expresses the opinion that, to be mature for publication, our paper should cover a broader area to highlight the impact of no crowd allowance in more sports or leagues.

We fully agree with the reviewer that, in order to generalize our results, the coverage of a broader area is desirable. Since the issue has already been studied in other sports (see references 30, 35-39) and some sports are more difficult to evaluate (e.g., baseball where referees do not call penalties or American football where crowd noise has been added in empty stadiums), we decided to apply our model to other leagues. 

We first tried to find another professional ice hockey league that held playoffs during the pandemic. Our search left us empty-handed, as the professional leagues have all cancelled their 2020 playoffs (with the exception of the NHL of course) and the 2021 playoff formats offers little or no usable data. Indeed, many leagues held their 2021 series in front of small crowds (Kontinental Hockey League, Czech Extraliga, Slovakia Extraliga), others with a format generating less than 40 games in empty arenas (Finland Liiga, Swiss National Hockey League, Swedish Hockey League, Deutsche Eishockey Liga), while the other leagues simply cancelled their 2021 playoffs again (American Hockey League, Men’s England Hockey League, Ligue Magnus de France). Afterwards, we looked for junior hockey leagues that might have offered more interesting opportunities. The Canadian Hockey League has played the most games in empty arenas, with 44, which is still too small to benefit from sufficient statistical power. The lack of overall opportunities led us to turn to the regular season.

Since we had to use regular season statistics, we decided to add the NHL season statistics that were not part of the precedent version of this manuscript. Because the stakes are different in the season than in the playoffs, the simple difference in the importance of the games can have an impact on the NHL referees' behavior. This is indeed what we found, with a slightly lower variance explained by our model in the regular season. We then decided to include a different level league, to try to see if the referees are also influenced by the crowd. The Canadian Hockey League, which includes 3 leagues and offers many opportunities to collect games without spectators, was chosen. Played mostly in Canada, where health restrictions have prevented fans from attending games for a long time, this league allows us to collect a lot of data. In addition, it represents a lower-level league than the NHL, which allows the generalization of results to other levels of competition. Moreover, this league is known for having passionate fans who are involved with their local team, which means that even small crowds could generate a lot of social pressure on the referees.

We were able to replicate the results in the different conditions and leagues. Considering the addition of regular season data in the NHL and CHL, several changes have been made to the article to explain our approach, present the results, and discuss them. The changes to the manuscript appear as follows:

Abstract. We have added the following text to lines 30-35:

“In order to generalize these results, we took advantage of the extension of the pandemic and the unusual game setting it provided to observe the behavior of referees during the 2020-2021 regular season. Again, using data from the National Hockey League (n=1639), but also expanding our sample to include Canadian Hockey League games (n=1709), we also find that the advantage given to the home team by referees when in front of a crowd fades in the absence of spectators.”

Introduction. We have added the following text to lines 105-114:

“. To ensure the generalizability of the results, analysis of referee behavior in various contexts is also presented. This was done by comparing 2020-2021 regular season games from the NHL, as well as from the Canadian Hockey League (CHL), the top junior ice hockey league in Canada, to games from their respective previous regular seasons. Those additional data sets offer the possibility to observe penalties awarded by referees in different contexts, when compared to the initial sample. Notably, the new data sets looked at games in which the stakes are minimized compared to the playoffs, at games played in the teams' local arenas rather than in neutral hub cities, and at games where the playing level and the degree of expertise of the referees are lower.”

Materials and methods. Results. We changed the section "Materials and Methods" to "Methods and Results". We were then able to include the following subheadings to better illustrate the different parts of our project: NHL playoffs data set; NHL playoffs model; NHL playoffs analysis and results; Regular season data sets; Regular seasons model; Regular seasons analysis and results. The first 3 sub-sections contain only text that was already in the first manuscript. The other sections are as follows, from page 11 to page 16:

Regular season data sets

To investigate the generalizability of our results, we used the same strategy and variables as for the NHL playoff data set to construct additional data sets for the NHL and CHL regular seasons. First, we aimed to see if crowd pressure had an impact on referee behavior in games where the stakes are lower and where games are played in the local teams' arenas rather than in neutral hub cities. Secondly, we wanted to test whether the influence of spectators was observable on referee decision-making in a league below the NHL level. The collection of data from games played in empty arenas was made possible by the extension of restrictive health measures related to the COVID-19 pandemic that forced some hockey leagues to begin the 2020-2021 season without crowds. This is the case for the NHL and CHL. In addition to having to play some games in front of empty seats, these leagues have also been compelled to shorten their regular schedule to avoid cancelling their seasons altogether. For the NHL, we once again used the official website nhl.com to collect the data. As for the Canadian Hockey League, we used the subdivisions of the official website (chl.ca) to acquire game summaries from the individual leagues that make up the CHL (whl.ca, ohl.ca, theqmjhl.ca).

We started by collecting data for the 2020-2021 NHL season, in which teams were schedule to play 56 games instead of the usual 82 games that are normally played in non-pandemic regular season. Unlike the playoffs, no hub city was designated, and teams played in their respective hometown. This is closer to the usual format of NHL games, providing an opportunity to verify that the 2020 playoff results are not due to hub cities being neutral ground. The NHL’s four divisions were reformatted with the goal of having teams exclusively play against other teams in their division. Given that government restrictions prohibited border crossings between the Canada and the United States, an exclusively Canadian division was created, along with three American divisions. All the games in the Canadian division were played in empty arenas. For the American division games, state-specific rules dictated whether or not spectators were present, with some games played in front of limited crowds (n=310). These games were removed from the sample because the very limited number of fans might not generate the same pressure on the referees as in an arena full of spectators. Nevertheless, additional statistical analyses that include these games were performed to confirm that removing them does not change the results. 

Next, we compiled the data for the 2020-2021 CHL season. The CHL is made up of three leagues spread across Canada and the United States. The Western Hockey League (WHL), the Ontario Hockey League (OHL) and the Quebec Major Junior Hockey League (QMJHL) includes teams from nine provinces of Canada and four American states. As public health measures vary between provinces and states, the structure of the regular schedule also varied between the 3 leagues that make up the CHL. For starters, health measures caused the OHL to completely cancel its 2020-2021 season. For their part, the WHL and QJMHL opted for a shorter schedule. Since the teams that make up these two leagues are spread across several Canadian provinces and some American states, the format of the games varied according to the public health measures of each province and state. As a result, some of the games in the QJMHL were played in front of fans (n=102). These games were removed from the sample. Once again, additional statistical analysis including the data from these games confirms that their removal does not alter the significance level of the results. Public health restrictions also led both leagues to hold some games in hub cities, similar to the NHL during its 2020 playoffs, while the rest of the games were held in the respective teams' hometowns.

In order to compare decisions taken by referees in this unusual context with decisions taken by referees during typical games played in front of crowds, we used the entirety of the previous season's games (2019-2020 regular season) for both the NHL and CHL. For the CHL, we excluded OHL games since it did not play games in 2020-2021 and meant it could not be used for comparison purposes. Although the 2019-2020 seasons were somewhat shortened due to the outbreak of the pandemic, the number of games in each league is far greater than the number of games suggested by Lakens and Evers [25] to observe a small effect size. Descriptive statistics for the regular seasons are presented in Table 4.

Table 4. Penalty kills per year during the NHL and CHL regular seasons.

Year Games Crowd Away PK Home PK Mean Away PK Mean Home PK

NHL 

 2019-2020 1082 Yes 3368 3064 3.11 (1.484) 2.83 (1.353)

 2020-2021 557 No 1634 1643 2.93 (1.484) 2.95 (1.452)

CHL 

 2019-2020 1264 Yes 4979 4651 3.94 (1.791) 3.68 (1.648)

 2020-2021 445 No 1660 1713 3.73 (1.727) 3.85 (1.830)

Standard deviations are presented in parentheses.

Regular seasons model

The same Poisson GLMM was used to estimate the number of penalties called by NHL and CHL referees during regular seasons. The model for both data sets still meets assumption of equidispersion and data were not zero inflated or zero truncated [28,29]. The R packages rsq [32] and emmeans [33] were again used to estimate the adjusted R2 and to run post-hoc analyses, respectively.

Regular seasons analysis and results

The model illustrates a significant effect of the Home*Crowd interaction for NHL regular season games (b = .10, p= .010) and for CHL regular season games (b = .10, p= .018). The results suggest that the probability of penalties being called by referees in these two leagues is modified by team status and the presence or absence of fans. The Poisson GLMM results for both leagues in regular season are presented in Table 5. 

Table 5. Estimated regression parameters for NHL (n=3278) and CHL (n=3418) regular seasons Poisson GLMM. 

 Estimate

(b) Std. error z value P-value IRR

Exp(b)

NHL 

 Intercept 1.033 0.111 9.437 <0.001 2.851

 Home -0.093 0.022 -4.179 <0.001 0.911

 Crowd -0.068 0.040 -0.114 0.909 0.995

 Home : Crowd 0.101 0.043 2.571 0.010 1.116

 Referees FE Yes

 Team FE Yes

 Adjusted R2 0.081

CHL 

 Intercept 0.320 0.580 0.552 0.581 1.377

 Home -0.066 0.020 -3.251 0.001 0.936

 Crowd -0.035 0.034 -1.030 0.303 0.966

 Home : Crowd 0.095 0.040 2.362 0.018 1.099

 Referees FE Yes

 Team FE Yes

 Adjusted R2 0.077

Estimates are in log-odds from a Poisson regression

FE: Fixed effect

IRR: Incidence rate ratio

Simultaneous pairwise comparisons using Tukey's HSD test indicated that in front of a crowd, penalties awarded to the away team were significantly higher than those awarded to the home team in both the NHL (Z = 3.729, p=.001) and CHL (Z = 3.251, p=.006). Similarly to the 2020 NHL playoffs, these differences fade in regular season games where arenas are empty in both the NHL (Z = -.230, p=.996) and the CHL (Z = -.820, p=.845). In contrast to the NHL playoffs, no other significant differences are observed in the regular season in either the NHL or the CHL. The model for NHL explains 8.1% of the variance, while the model for CHL measures 7.7% of the variance. Again, a small effect size is observable, reinforcing the idea that external factors generate only a small influence on referee penalty calls. 

Table 6. Post-hoc results of the NHL (n=3278) and CHL (n=3418) models during regular seasons.

 Estimate

(b) Std. error z ratio P-value IRR

Exp(b)

NHL 

 Away Crowd – Away Empty 0.068 0.031 2.181 0.129 1.070

 Away Crowd – Home Crowd 0.093 0.025 3.729 0.001 1.098

 Away Crowd – Home Empty 0.060 0.032 1.899 0.228 1.062

 Away Empty – Home Crowd 0.025 0.032 0.790 0.859 1.025

 Away Empty – Home Empty -0.008 0.035 -0.230 0.996 0.992

 Home Crowd – Home Empty -0.033 0.032 -1.037 0.728 0.967

CHL 

 Away Crowd – Away Empty 0.035 0.034 1.030 0.732 1.036

 Away Crowd – Home Crowd 0.066 0.020 3.251 0.006 1.069

 Away Crowd – Home Empty 0.007 0.034 0.199 0.997 1.007

 Away Empty – Home Crowd 0.031 0.034 0.918 0.795 1.032

 Away Empty – Home Empty -0.028 0.035 -0.820 0.845 0.972

 Home Crowd – Home Empty -0.060 0.034 -1.755 0.236 0.942

Results are averaged over the levels of: Ref1, Ref2, Team 

Estimates are in log-odds 

Fig 2 and Fig 3 present violin plots with mean penalty kills (±1 SD error bars) for away and home teams, in front of crowds before the pandemic and in empty arenas during the 2020-2021 regular seasons. 

Fig 2. Mean Penalty kills per team with and without attendance during the NHL regular season. For the sake of simplicity, the raw data have been presented in the figure, even though the model results in Table 5 and Table 6 are in log-odds.

Fig 3. Mean Penalty kills per team with and without attendance during the CHL regular season. For the sake of simplicity, the raw data have been presented in the figure, even though the model results in Table 5 and Table 6 are in log-odds.

Consistent with the results stemming from the 2020 NHL playoffs, NHL and CHL referees, during regular season games, award the home team significantly fewer penalties on average compared to the away team. The results again corroborate our hypothesis that the home team is no longer favored when it comes to penalty calls in situations where there are no local fans in the arena.”

Discussion. Finally, the discussion has been modified to take into account the new results. The changes are as follows and are available on pages 17-20:

“Home advantage is a well-established concept in professional sport and the influence of the home crowd on referee decision-making is a factor that is assumed to contribute to this phenomenon. Using a unique ecological context generated by the COVID-19 pandemic, we found evidence of the effect of crowds on NHL referees’ decisions during the playoffs. Our observations can also be generalized to different contexts, i.e., regular season and lower-level league. These findings add to existing scientific evidence supporting the involvement of sports referees in home advantage [30,35-39] and reinforce the existence of this phenomenon in elite ice hockey. The results also allow us to better understand the decision-making processes of ice hockey referees, who have the potential to change the outcome of games.

Our findings provide new evidence linking referees’ decision-making with the concept of home advantage in elite ice hockey. More precisely, our model suggests that the home team receives less penalties when they play in front of their home crowd compared to the opposing team. This advantage disappears when the games are played without spectators. These results suggest that referees are more lenient towards home players when they are playing in front of their local fans. 

The social pressure exerted by fans on referees is a plausible explanation for their behavior [19-23]. Crowd pressure seems to have an impact on referees in different ice hockey leagues, as well as in competitive settings with varying stakes. However, the behavior of referees seems to be affected differently depending on the context. In the regular season, despite the difference in penalties called between the home and visiting teams when the crowd is present, it is difficult to determine whether the referees are calling more penalties on the visiting team, fewer penalties on the home team, or a combination of both. During the playoffs, a more specific direction of bias towards the home team is identifiable. Consistent with our results, crowd pressure during the playoffs can be observed by a decrease in the number of penalties awarded by the referees to the home team, a phenomenon that can be identified as an omission bias [40]. This bias would be more prevalent in critical moments, such as playoffs, where referees are known to try to limit their impact on the flow of the game [41]. Moreover, referees who display omission biases might be recognized as being fairer than referees who exhibit other types of biases. Indeed, the omission of penalties imposed on the home team can be considered less intrusive or damaging to the visiting team than an act of commission, such as an increase in the penalties imposed on the latter [41]. An excellent example of the perceived immorality of a commission call is the 2021 dismissal of Tim Peel, an NHL referee, who was caught admitting that he was trying to balance out the number of penalties awarded during the game by giving a questionable penalty to one of the teams [42]. To this day, there has never been a dismissal of an NHL referee who willingly omitted to call a penalty, or as they say in sports jargon, “swallowed his whistle” [41]. However, these examples raise the need to be cautious in interpreting an omission bias among NHL referees during the playoffs. While, as mentioned above, the greater stakes of playoff games may be sufficient to explain the omission bias, it is also important to consider that the referees selected to officiate these games are those whom the league considers their best. These referees can be perceived as being the best precisely because they tend to omit penalties to the home team, rather than awarding more penalties to the opposing team, which is considered more morally acceptable. In this case, the presence of the omission bias during the playoffs could be due to the selection of referees by NHL executives, rather than the higher stakes of the games being officiated. It would be relevant to assess this in future studies.

Given that the results are based on a unique ecological environment, the number of games played without spectators remains low compared to the data available for usual games played in front of a crowd. Also, since playoff games played without an audience are limited to the NHL in 2020, our findings in playoffs are not generalizable beyond a reasonable doubt to other ice hockey leagues. As far as we know, the NHL is the only league to have played playoffs in 2020. In 2021, the professional leagues that have held playoffs have done so with a small number of games or in front of a limited number of fans. In the CHL, only the QMJHL has held its playoffs in an adjusted format, limiting the number of games in empty arenas to 44, which does not provide enough statistical power for analysis. Also, our sample is limited to NHL and CHL referees, meaning that other referees in minor and major hockey leagues may behave differently. Furthermore, given that these leagues are played in North America, referees may be influenced differently than those in other continents. Since the style of refereeing may be influenced by cultural factors, it could be conceivable to expect different results from European referees, for example. Our project is limited to the NHL and CHL, and more research needs to be conducted in different leagues and countries before we can generalize our results to the broad population of ice hockey referees.

However, our study has the obvious strength of presenting an ecological design made possible by the COVID-19 pandemic. For the very first time in studies relating to elite referee behavior, the local crowd and its possible impact on decision-making was physically eliminated from the equation. This offers a significant advantage over designs where audience influence had to be statistically controlled. The statistical model used also adds strength to our results, whereas previous studies supporting the influence of the public on referees’ decision-making, mainly those made in soccer, have relied on ordinary linear regression models that does not fit the count data as well as Poisson regression [28]. Moreover, our study examined the behavior of referees working for the world’s best professional ice hockey league, as well as for one of the top junior hockey leagues. These referees are considered to be amongst the best in their discipline and the least likely to be influenced by factors outside of the game itself. The fact that our results present an impact of the crowd on elite referees’ decision-making suggests that this impact should also be present, and possibly greater, in less experienced or capable referees. Finally, our study exposes the effect of crowds on referee decisions in a variety of contexts. Social pressure from the crowd influences referees in more than one league, in leagues of different levels, as well as in games with different stakes. Our research project illustrates that this impact disappears with the absence of crowds, both in neutral hub cities and in hometown arenas.”

---

## [Decision Letter · Decision Letter 1]

10 Aug 2021

The absence of fans removes the home advantage associated with penalties called by National Hockey League referees

PONE-D-21-15462R1

Dear Dr. Fiset,

We’re pleased to inform you that your manuscript has been judged scientifically suitable for publication and will be formally accepted for publication once it meets all outstanding technical requirements.

Kind regards,

Haroldo V. Ribeiro

Academic Editor

PLOS ONE

Reviewers' comments:

Reviewer's Responses to Questions

**Comments to the Author**

1. If the authors have adequately addressed your comments raised in a previous round of review and you feel that this manuscript is now acceptable for publication, you may indicate that here to bypass the “Comments to the Author” section, enter your conflict of interest statement in the “Confidential to Editor” section, and submit your "Accept" recommendation.

Reviewer #1: All comments have been addressed

Reviewer #2: All comments have been addressed

2. Is the manuscript technically sound, and do the data support the conclusions?

Reviewer #1: (No Response)

Reviewer #2: Yes

3. Has the statistical analysis been performed appropriately and rigorously? 

Reviewer #1: (No Response)

Reviewer #2: Yes

4. Have the authors made all data underlying the findings in their manuscript fully available?

Reviewer #1: (No Response)

Reviewer #2: No

5. Is the manuscript presented in an intelligible fashion and written in standard English?

Reviewer #1: (No Response)

Reviewer #2: Yes

6. Review Comments to the Author

Reviewer #1: The authors are sufficiently answered to all of my questions. The paper is now suitable for publication.

Reviewer #2: (No Response)

7. PLOS authors have the option to publish the peer review history of their article (what does this mean?). If published, this will include your full peer review and any attached files.

Reviewer #1: No

Reviewer #2: No

---

## [Editor Report · Acceptance letter]

12 Aug 2021

PONE-D-21-15462R1 

The absence of fans removes the home advantage associated with penalties called by National Hockey League referees 

Dear Dr. Fiset:

I'm pleased to inform you that your manuscript has been deemed suitable for publication in PLOS ONE. Congratulations! Your manuscript is now with our production department. 

Kind regards, 

on behalf of

Dr. Haroldo V. Ribeiro 

Academic Editor

PLOS ONE